# Development and Validation of Quality Indicators for Pulmonary Arterial Hypertension Management in Japan: A Modified Delphi Consensus Study

**DOI:** 10.3390/diagnostics14232656

**Published:** 2024-11-25

**Authors:** Yuichi Tamura, Kazuya Hosokawa, Koshin Horimoto, Satoshi Ikeda, Takumi Inami, Kayoko Kubota, Naohiko Nakanishi, Yuichiro Shirai, Nobuhiro Tanabe, Ichizo Tsujino, Hiromi Matsubara

**Affiliations:** 1Pulmonary Hypertension Center, International University of Health and Welfare Mita Hospital, Tokyo 108-8329, Japan; 2Department of Cardiology, International University of Health and Welfare School of Medicine, Narita 286-8520, Japan; 3Faculty of Cardiovascular Medicine, Graduate School of Medical Sciences, Kyushu University, Fukuoka 812-8582, Japan; 4Center for Advanced Medical Innovation, Kyushu University, Fukuoka 812-8582, Japan; 5Department of Cardiology, Matsuyama Red Cross Hospital, Matsuyama 790-0826, Japan; 6Stroke and Cardiovascular Diseases Support Center, Nagasaki University Hospital, Nagasaki 852-8501, Japan; 7Department of Cardiovascular Medicine, Kyorin University School of Medicine, Tokyo 181-8611, Japan; 8Department of Cardiovascular Medicine and Hypertension, Graduate School of Medical and Dental Sciences, Kagoshima University, Kagoshima 890-0065, Japan; 9Department of Cardiovascular Medicine, Graduate School of Medical Science, Kyoto Prefectural University of Medicine, Kyoto 602-8566, Japan; 10Department of Allergy and Rheumatology, Nippon Medical School, Graduate School of Medicine, Tokyo 113-8602, Japan; 11Pulmonary Hypertension Center, Chibaken Saiseikai Narashino Hospital, Chiba 275-8580, Japan; 12Division of Respiratory and Cardiovascular Innovative Research, Faculty of Medicine, Hokkaido University, Sapporo 060-8638, Japan; 13Department of Cardiology, NHO Okayama Medical Center, Okayama 701-1192, Japan; matsubara.hiromi@gmail.com

**Keywords:** pulmonary arterial hypertension, quality indicator, expert consensus, PH centers

## Abstract

Background: Quality indicators (QIs) are used to standardize care and improve outcomes in patients with pulmonary arterial hypertension (PAH). It is important that QIs are validated within specific healthcare contexts. Therefore, this study aimed to validate QIs for PAH management in Japan using a modified Delphi consensus method. Methods: QI candidates were identified from published European QIs and clinical practice guidelines. An expert panel of 11 PAH specialists from diverse Japanese institutions anonymously rated the 36 initial QI candidates in two rounds using a nine-point appropriateness scale. Results: In the first round, 35 QIs received a median score of ≥7 points. A panel discussion was held between rounds to address the single low-scored QI, biomarker modifications, and invasive examinations, resulting in 36 modified QIs. In the second round, all modified QIs received median scores of ≥7 points and were judged to be valid as the final Japanese set of QIs. Conclusions: The findings of this study validated a set of QIs for PAH management tailored to the Japanese healthcare context. These QIs can be used to standardize care, identify areas for improvement, and ultimately enhance outcomes for Japanese patients with PAH.

## 1. Introduction

Pulmonary arterial hypertension (PAH) is a severe and progressive disease, defined by increased pulmonary artery pressure and vascular resistance, which can progressively cause right ventricular dysfunction and potentially result in mortality [1]. Despite advances in pharmacological therapies, PAH remains a challenging condition with a substantial burden on healthcare systems worldwide.

The European Society of Cardiology (ESC) and the European Respiratory Society (ERS) have developed clinical practice guidelines that provide evidence-based recommendations for the diagnosis, treatment, and management of PAH [2]. These guidelines aim to translate scientific knowledge into clinical practice. However, the implementation of guideline-recommended therapies for PAH remains suboptimal, leading to disparities in clinical outcomes across different regions and healthcare settings [3,4].

In this context, the development and application of quality indicators (QIs) can help standardize the structure and processes of care, benchmark clinical practices against established measures, and improve patient outcomes. QIs provide a framework for healthcare professionals to evaluate their clinical practices, identify areas for improvement, and implement targeted interventions to enhance the quality of care. Several countries have recognized the necessity of adapting QIs to their specific healthcare contexts. For instance, in the United States, the American College of Cardiology (ACC) and the American Heart Association (AHA) have developed localized QIs to improve cardiovascular care [5]. Similarly, in Canada, QIs have been tailored to align with national guidelines and healthcare policies [6]. These adaptations ensure that QIs are culturally relevant, feasible, and effective within different healthcare systems. By learning from these international efforts, we aim to adapt and validate QIs for PAH management that are specifically suited to the Japanese context.

Aktaa et al. [7] emphasized the importance of QIs in PAH management to help healthcare professionals benchmark clinical practice against standardized measures to identify areas for improvement. They also highlighted the importance of addressing potential inequalities and improving patient experience through the standardization of PAH care and the systematic capture of outcomes. Although QIs for pulmonary hypertension management have been proposed in other regions, it is desirable to develop QIs based on the latest European guidelines, which are globally recognized and referenced [7]. Adapting these QIs to reflect the actual clinical practice in Japan is crucial, as variations in healthcare systems, disease prevalence, and practice patterns can significantly impact their applicability and effectiveness.

Validating PAH management QIs specific to the Japanese context is a crucial step toward enhancing the quality of care for patients with PAH in Japan. By establishing a set of validated QIs, healthcare professionals can assess their adherence to evidence-based practices, identify gaps in care delivery, and implement targeted interventions to improve patient outcomes and reduce disparities in PAH management.

## 2. Materials and Methods

### 2.1. QI Selection and Adaptation

We conducted a study to validate QIs for the clinical management of adult PAH in Japan, adapting the Research and Development/University of California at Los Angeles (RAND/UCLA) Appropriateness Method (Figure 1) [8].

This method combines the best available scientific evidence with the collective judgment of experts to produce criteria for appropriate clinical practices through a modified Delphi process.

We began with the 52 QIs developed by Aktaa et al. [7], which are based on the ESC/ERS guidelines. Three PAH experts independently reviewed each QI for relevance and applicability to the Japanese healthcare context. The review considered factors such as the prevalence of certain PAH subtypes in Japan, availability of diagnostic tools, and standard treatment practices. We applied specific criteria to assess each QI’s relevance and applicability to the Japanese context. These criteria included the relevance to Japanese clinical practice, ensuring that QIs reflected common practices in Japan; feasibility, considering the ability of healthcare providers to implement the QI given available resources and technology; alignment with the most current and robust clinical evidence; and compatibility with Japan’s healthcare policies and cultural norms. QIs that did not meet these criteria were modified or excluded. For example, certain diagnostic procedures not routinely used in Japan were adjusted to reflect standard practice.

QIs that did not meet these criteria were modified or excluded. For example, certain diagnostic procedures not routinely used in Japan were adjusted to reflect standard practice.

### 2.2. Expert Panel Selection and Composition

The Delphi method recommends a panel size that balances diversity with manageability, typically ranging from 10 to 18 experts. We selected 11 panelists to ensure a broad representation of expertise while maintaining an efficient consensus process. Each panelist is a recognized leader in PAH management, contributing extensive clinical experience and academic insight. Experts were selected based on the following criteria: Each expert had at least 10 years of experience in the management of PAH patients and had contributed to the Japanese PAH registry. The panel comprised 11 specialists: 5 cardiologists, 4 pulmonologists, and 2 rheumatologists. This multidisciplinary team reflects the collaborative nature of PAH management and ensures that the QIs are applicable across various clinical settings. Each panelist brought unique expertise to the evaluation process. Cardiologists provided insights on cardiovascular assessments and interventions specific to PAH; pulmonologists offered expertise on respiratory function tests and pulmonary imaging; and rheumatologists contributed knowledge on connective tissue diseases associated with PAH. This multidisciplinary expertise ensured a comprehensive evaluation of each QI from various clinical perspectives, enhancing the validity of our consensus. This multidisciplinary expertise ensured a comprehensive evaluation of each QI from various clinical perspectives, enhancing the validity of our consensus.

### 2.3. Rating Process (Round 1)

Each QI candidate was shared with the panel members via a web-based system (Google Forms). The panel members individually and anonymously reviewed the QI candidates and rated their appropriateness for the Japanese clinical setting using a nine-point scale, where 1 represented “definitely inappropriate” and 9 represented “definitely appropriate” (Round 1 rating).

### 2.4. Panel Discussion and Re-Rating (Round 2)

Subsequently, an online panel meeting was held to discuss the initial ratings collectively. Referring to the summarized results of the Round 1 ratings, the panel members engaged in a structured discussion, critically evaluating the QI candidates and their relevance to the Japanese healthcare system. During this discussion, panel members had the opportunity to provide additional insights, raise concerns, and share their perspectives on each QI candidate, with a focus on items with low scores or significant comments. The nine-point appropriateness scale is a fundamental aspect of the RAND/UCLA Appropriateness Method, chosen for its ability to capture nuanced expert opinions on the suitability of clinical practices [8]. This scale allows panelists to express varying degrees of agreement or disagreement, facilitating a more precise consensus. Its use is justified by its widespread acceptance and effectiveness in similar studies assessing clinical appropriateness.

### 2.5. Final QI Selection

Only those QI candidates that were rated as appropriate (median score ≥7 points) by most panelists, with minimal disagreement, were adopted as the final set of validated QIs for PAH management in Japan. We assessed the consensus level utilizing the RAND/UCLA Appropriateness Method criteria [8], factoring in both the median scores and the variability among panelists’ ratings. According to these criteria, indications with median scores of 1–3 are considered as inappropriate, scores of 4–6 as uncertain, and scores of 7–9 as appropriate.

## 3. Results

### 3.1. Development of the Japanese Provisional Version of the QIs

First, based on the English version of the indicators reported by Aktaa et al. [7], a Japanese version was created. The linguistic validation was conducted by independent professional translators who were not study authors, ensuring unbiased translation and back-translation processes. The Japanese version of the QIs underwent reconciliation of discrepancies and cognitive debriefing with native Japanese translators to verify clarity, cultural appropriateness, and conceptual equivalence of the translated scale. After undergoing this appropriate translation process, three experts in pulmonary hypertension management were involved in creating a list of indicators suitable for the Japanese version, which was developed as a provisional Japanese version. This provisional version consisted of 36 question items across five domains, similar to that of Aktaa et al. [7] (Appendix A).

### 3.2. Round 1 Rating Process

A panel consisting of 11 experts in PAH management from across Japan was formed, which included pulmonologists, cardiologists, and rheumatologists with extensive experience in PAH care and research. The affiliated institutions of these 11 members ranged from large-scale pulmonary hypertension treatment centers to regional hospitals. After providing a 15 min web-based video explanation of the procedure to the selected expert panel, the committee members provided individual anonymous evaluations. The results of the evaluation are shown in Figure 2A and Appendix A, with 35 of the 36 question indicators having a median score of ≥7 points, with most being 9 points.

### 3.3. Panel Discussion

The panel discussion included 10 expert panelists, who determined whether to adopt or reject the “Main 3.3 Proportion of patients with a diagnosis of vasoreactive idiopathic, heritable, or drug-associated PAH and acute vasodilator response who are prescribed high doses of calcium channel blockers” indicator, which was the only one with a score of 6 points. The main concern raised by the panelists was that, compared with reports from Europe and the United States, the proportion of vasoreactive cases is low in Japan, and many medical facilities rarely encounter such cases in practice.

Considering the guideline recommendations, the ability to perform right heart catheterization in patients at high risk of developing PAH during diagnostic workup was considered important. However, as the significance of therapeutic interventions for exercise-induced PAH cases has not been established, this was adopted as a secondary indicator. These points were proposed by the cardiologists in the group.

In addition, for other items that some panelists rated in the 1–3-point range, the panel Discussion suggested that consensus may not have been reached for those items. The QIs 2.7 and 4.3 originally specified NT-proBNP for patient assessment. However, in Japan, BNP is preferred over NT-proBNP for its accessibility. Based on this clinical practice pattern, the panel modified these QIs to include both options as “NT-proBNP (or BNP).” This change was also proposed by cardiologists to better align with Japanese medical practice.

### 3.4. Re-Rating (Round 2) and Final QI Selection

After review by the panel discussion, a revised version of the QI indicators was constructed (Table 1, the Japanese version is shown in Appendix A).

The same 11 panelists participated in the Round 2 rating process, the results of which confirmed that all items had a median score of ≥7 points and were determined to comprise the final version of the QIs (Table 1, Figure 2B).

## 4. Discussion

This study developed a set of validated QIs for the clinical management of PAH in Japan, which were primarily adapted from European QIs [7]. The adaptation process involved a panel of national experts and utilized the RAND/UCLA Appropriateness Method [8], which combines scientific evidence with expert clinical judgment to ensure the QIs’ relevance and applicability to the Japanese healthcare context.

A significant number of the original European QIs were adopted for the Japanese context, reflecting the universal nature of evidence-based practices and guideline recommendations. This alignment underscores the global consensus on key aspects of PAH management, such as timely diagnosis, appropriate use of therapies, and regular patient monitoring [9]. However, the expert panel made necessary modifications, emphasizing the need to customize QIs to the unique clinical and healthcare systems of different countries. Specifically, the evaluation periods or frames of some QIs were adjusted to better suit the practicalities and feasibility of implementation in Japanese clinical settings. For instance, the incorporation of biomarkers like BNP as well as NT-proBNP acknowledges differences in routine clinical practices and available resources [10]. This approach to localization is supported by previous research [11,12], which highlights that adapting guidelines and QIs to the local context can enhance their acceptance and effectiveness [13].

One of the pivotal QIs we validated is the use of right heart catheterization (RHC) at the time of diagnostic work-up (QI 2.3). RHC is essential because it provides definitive hemodynamic measurements necessary for diagnosing PAH, assessing severity, and guiding treatment decisions [14]. Its importance in the diagnostic algorithm cannot be overstated, as misdiagnosis can lead to inappropriate management.

The developed QIs offer a potential framework for healthcare professionals and institutions managing PAH in Japan. Comparison of clinical practices against these standards can identify areas for improvement, allow the implementation of targeted interventions, and enhance the quality of care of patients with PAH. Moreover, implementing these QIs can facilitate benchmarking across institutions, promoting a collaborative environment for shared learning and quality improvement initiatives [15]. This process of continuous quality improvement promotes a culture of self-evaluation and refinement, which will improve patient outcomes in the cardiovascular field [16].

Moreover, these Japanese PAH management QIs could guide future revisions of national clinical practice guidelines. Integrating these QIs into guidelines helps to standardize care and promote the adoption of evidence-based practices across healthcare facilities, thus reducing disparities in PAH management and ensuring consistent, high-quality patient care nationwide. Standardization is particularly crucial in rare diseases like PAH, where variations in care can significantly impact patient outcomes [14,17].

This study has several limitations. The relatively small panel size, while sufficient for the Delphi method, may not capture all regional or institutional variations in PAH management across Japan. Additionally, although efforts were made to include a diverse group of experts, some specialties or regions may be underrepresented. Furthermore, the reliance on expert opinion may introduce bias, and the practical implementation of these QIs remains to be evaluated in clinical settings [18]. Furthermore, although our expert panel was multidisciplinary and included nationally recognized specialists, it may not encompass the full spectrum of PAH management practices across Japan. In particular, practices in rural or less-resourced settings might differ, potentially affecting the generalizability of the validated QIs to all healthcare environments within the country. Additionally, our study did not involve input from patients or other healthcare professionals such as nurses, pharmacists, and social workers who play integral roles in PAH management. Including these stakeholders could have provided a more holistic view and enriched the consensus process. Future research should aim to incorporate a broader range of perspectives to enhance the relevance and acceptance of the QIs.

Future research should aim to validate these QIs with a larger and more diverse group of stakeholders, including patients and other healthcare professionals involved in PAH care. Considering the variations in healthcare resources and practices across different regions and institutions in Japan, assessing the feasibility of implementing these QIs is crucial. Conducting pilot studies or practical implementation testing in a range of healthcare settings would provide valuable insights into the effectiveness, practicality, and potential challenges of adopting the QIs nationwide. This approach would strengthen confidence in the indicators and guide necessary adaptations for broader applicability. Assessing the impact of these QIs on clinical outcomes and patient satisfaction will be essential to determine their effectiveness and guide further refinements [19]. To ensure that the QIs continue to reflect best practices, it is essential to establish mechanisms for regular feedback, monitoring, and updates. Finally, future development and validation of QIs should involve a broader range of stakeholders, including patients and various healthcare professionals involved in PAH care. Engaging these groups would provide diverse perspectives, enhance the relevance and acceptability of the QIs, and potentially improve patient outcomes through more holistic care strategies.

## 5. Conclusions

This study established a robust set of validated QIs for PAH management in Japan, which are tailored to local clinical needs while aligned with international standards. Implementing these QIs can standardize care, improve clinical practices, and enhance outcomes for patients with PAH, thus contributing to global efforts to optimize care for this complex condition.

## Figures and Tables

**Figure 1 diagnostics-14-02656-f001:**
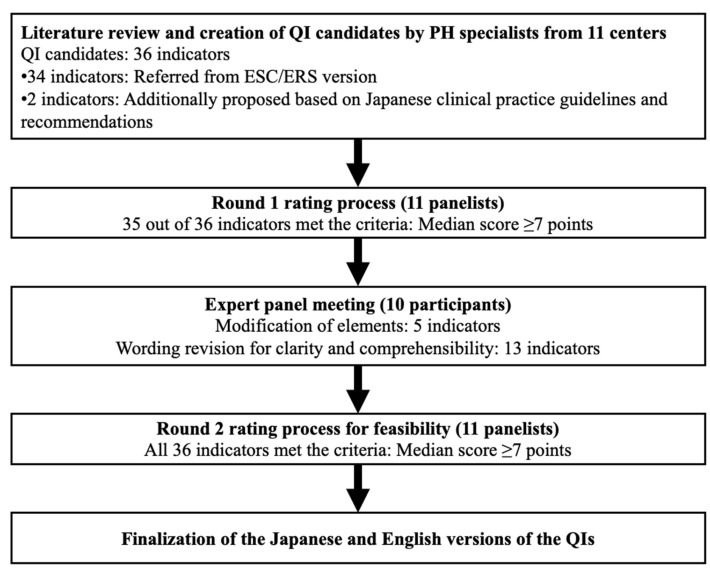
Overview of the modified Delphi process for developing quality indicators (QIs) for pulmonary arterial hypertension (PAH) management in Japan. The process included a literature review and creation of QI candidates by PAH specialists from 11 centers, resulting in 36 initial QI candidates (34 from European Society of Cardiology and European Respiratory Society guidelines and two proposed QIs based on Japanese clinical practice). The Round 1 rating process involved 11 panelists, in which 35 of 36 prospective QIs met the criteria (median score ≥ 7 points). An expert panel meeting with 10 participants led to modifications in five indicators and wording revisions for 13 indicators. The Round 2 (re-)rating process confirmed all 36 QIs as meeting the criteria (median score ≥ 7 points). The final versions of the QIs were completed in both Japanese and English.

**Figure 2 diagnostics-14-02656-f002:**
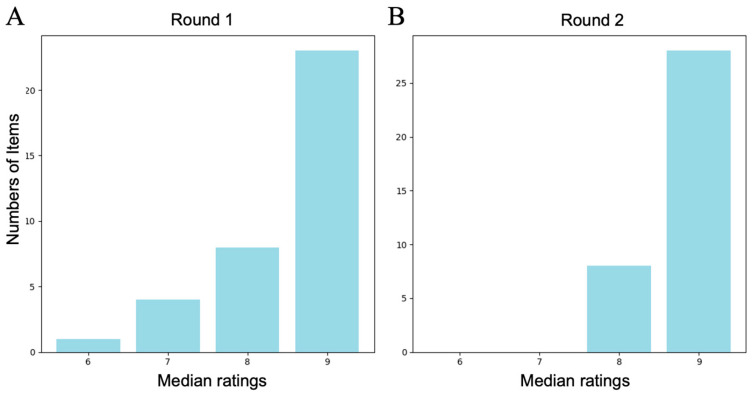
Results of the quality indicator (QI) rating process (*n* = 11 panelists). (**A**) Distribution of the panelists’ median ratings for each of the 36 proposed QIs in Round 1, with most indicators receiving a median score of 9 points. (**B**) Distribution of the panelists’ median ratings for each of the 36 QIs in Round 2, which confirmed that all indicators met the criteria (median score ≥ 7 points) and were validated as the final set of QIs for PAH management in Japan.

**Table 1 diagnostics-14-02656-t001:** Finalized set of the quality indicators for the management and outcomes of adults with pulmonary arterial hypertension. (Originally derived from [7].)

Domain	Content	**Med**	**Min**	**Max**	**Mo**
1. Structural framework					
**1.1**	Pulmonary hypertension centers that have a specialized MDT responsible for the management of patients with PAH **Note:** MDT consists of at least a cardiologist, pulmonologist, and specialist nurse. Collaboration should be established with a rheumatologist, interventional radiologist, cardiothoracic surgeon, social worker, and psychologist	9	8	9	9
**1.2**	Pulmonary hypertension centers that have the following facilities and skills:				
	- A ward where healthcare providers have expertise in PAH;	9	8	9	9
	- A specialist outpatient service;	9	7	9	9
	- An intermediate/intensive care unit;	9	8	9	9
	- A 24/7 emergency care;	9	8	9	9
	- An interventional radiology unit (for treatment of hemoptysis);	8	7	8	8
	- Diagnostic investigations, including echocardiography, CT scanning, nuclear medicine, MRI, exercise tests, and PFT;	9	8	9	9
	- A cardiac catheterization laboratory with vasodilator testing available;	9	4	9	9
	- Access to genetic counseling and testing;	9	7	9	9
	- Fast and easy access to cardiothoracic and vascular surgery, cardiac anesthesia, and ECMO;	9	8	9	9
	- Established collaboration with a lung/heart–lung transplantation center;	9	7	9	9
**1.3**	Pulmonary hypertension centers that participate in a national or an international PAH registry	9	7	9	9
**1.4**	Pulmonary hypertension centers that have a fast-track policy to review urgent referrals within 1–2 weeks	9	8	9	9
**2. Diagnosis and risk stratification**					
**2.1**	Proportion of patients with suspected PAH who undergo pulmonary function test (including lung volumes and DLCO) at the time of diagnostic work-up	9	8	9	9
**2.2**	Proportion of patients with suspected PAH who have an echocardiography at the time of diagnostic work-up	9	9	9	9
**2.3**	Proportion of patients with suspected PAH who have a RHC at the time of diagnostic work-up	9	8	9	9
**2.4**	Proportion of patients with suspected PAH who have perfusion imaging (V/Q scan or new modality) to exclude CTEPH at the time of diagnostic work-up **Note:** Alternative perfusion imaging techniques include iodine subtraction mapping, dual-energy CT, or MRI perfusion	9	9	9	9
**2.5**	Proportion of patients with suspected PAH who have been screened for CTD at the time of diagnostic work-up	9	8	9	9
**2.6**	Proportion of patients with a diagnosis of idiopathic, heritable, or drug-induced PAH who have RHC with acute vasodilator testing at the time of diagnostic work-up	9	4	9	9
**2.7**	Proportion of patients with a diagnosis of PAH who have their WHO-FC, NT-proBNP (or BNP) and 6MWT assessed at the time of PAH diagnosis	9	8	9	9
**2.8**	Proportion of patients with a diagnosis of PAH who have their risk assessed using a validated tool (e.g., ESC/ERS guidelines) at the time of PAH diagnosis	9	6	9	9
**2.9**	Proportion of patients with a diagnosis of PAH who have their quality of life assessed using a validated tool (e.g., Emphasis-10, SF-36, etc.) at the time of PAH diagnosis	8	5	9	8
**Secondary 2**	Pulmonary hypertension centers that can perform exercise RHC in patients with suspected PAH at high risk at the time of diagnostic work-up	8	4	9	8
**3. Initial treatment**					
**3.1**	Proportion of patients with a diagnosis of non-vasoreactive idiopathic, heritable, or drug-associated PAH and at high risk without significant cardiopulmonary comorbidities who are prescribed i.v./s.c. prostacyclin analogues	9	8	9	9
**3.2**	Proportion of patients with a diagnosis of non-vasoreactive idiopathic, heritable, drug-associated or CTD-associated PAH and at low or intermediate risk without significant cardiopulmonary comorbidities who are prescribed initial combination therapy with a NO donor and an ERA	9	8	9	9
**3.3**	Proportion of patients with a diagnosis of vasoreactive idiopathic, heritable, or drug-associated PAH and acute vasodilator response who are prescribed high doses of calcium channel blockers	8	4	9	9
**4. Follow-up**					
**4.1**	Proportion of patients with a diagnosis of PAH who have their risk assessed using a validated tool (e.g., ESC/ERS guidelines) at least every 6 months	9	6	9	9
**4.2**	Proportion of patients with a diagnosis of PAH who have been informed about available patient association/support group(s)	8	4	9	9
**4.3**	Proportion of patients with a diagnosis of PAH who have their WHO-FC, NT-proBNP (or BNP) and 6MWT assessed at least every 6 months	9	7	9	9
**4.4**	Proportion of patients with a diagnosis of PAH in whom low risk is not achieved who have a discussion with a member of the MDT on treatment strategy	8	7	9	9
**4.5**	Proportion of patients with a diagnosis of PAH and at intermediate-high or high risk who are evaluated for lung transplantation Note: Who are eligible for lung transplantation (based on age and comorbidities) and have been established on a combination therapy.	8	5	9	9
**4.6**	Proportion of patients with a diagnosis of PAH who have not achieved low risk for whom regular hemodynamic assessment is considered at least every 12 months	9	8	9	9
**Secondary 4**	Proportion of patients with a diagnosis of PAH who have their quality of life assessed using a validated tool at least every 6 months	8	6	9	8
**5. Outcomes**					
**5.1**	Median time between establishing the diagnosis of PAH (i.e., date of diagnostic RHC) and commencing PAH therapy	9	8	9	9
**5.2**	Median time between referral and commencing PAH therapy **Note:** Referral time is date of receipt of the referral request by the specialist PAH center	9	8	9	9

Med: median, Min: minimum, Max: maximum, Mo: mode, PAH: pulmonary arterial hypertension, MDT: multidisciplinary team, CT: computed tomography, MRI: magnetic resonance imaging, PFT: pulmonary function test, ECMO: extracorporeal membrane oxygenation, RHC: right heart catheterization, CTEPH: chronic thromboembolic pulmonary hypertension, DLCO: diffusing capacity of the lung for carbon monoxide, CTD: connective tissue disease, WHO-FC: World Health Organization functional class, NT-proBNP: N-terminal pro B-type natriuretic peptide, BNP: B-type natriuretic peptide, 6MWT: 6-Minute Walk Test, ESC/ERS: European Society of Cardiology/European Respiratory Society, SF-36, 36-Item Short Form Health Survey, i.v.: intravenous, s.c.: subcutaneous, NO: nitric oxide, ERA: endothelin receptor antagonist.

## Data Availability

The data presented in this study are available on request from the corresponding author.

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
