# Peer review of "Development and Validation of Quality Indicators for Pulmonary Arterial Hypertension Management in Japan: A Modified Delphi Consensus Study"

_diagnostics, 2024, doi:10.3390/diagnostics14232656_

Round 1
Reviewer 1 Report
Comments and Suggestions for Authors
Thank you for submitting this manuscript to the journal. This study has strong potential to enhance PAH care in Japan and set a model for other regions. However, I have couple of comments/points that need to be addressed first:
-
Introduction and Background
The introduction provides useful context, but it could be strengthened by including examples of previous adaptations of quality indicators (QIs) for specific countries or healthcare systems. This would reinforce the rationale for adapting European QIs specifically for Japan. For instance, discussing similar efforts in other regions would show the relevance and applicability of your approach. -
Methodology
Your choice of the RAND/UCLA Appropriateness Method and Delphi consensus process is appropriate and effective. However, the manuscript would benefit from clarifying the specific criteria used to modify or exclude certain QIs from the original European set. Additionally, a brief description of how each panelist’s expertise contributed to their evaluation would add depth and improve transparency in the methods section. -
Results Presentation
While the results are detailed, presenting each QI’s median, minimum, and maximum scores in a summary table would improve clarity and readability. This would give readers a clearer view of the consensus levels for each QI, making the findings more accessible and easier to interpret. -
Discussion and Implications
The discussion could be expanded to include mechanistic reasoning behind the relevance of each QI for PAH care. For instance, explaining why right heart catheterization is crucial in PAH management could make the indicators more relatable. Additionally, consider mentioning the potential for this localized approach to be adapted by other countries with specific healthcare needs, which would broaden the paper's impact. -
Limitations and Future Directions
You acknowledge the small panel size and reliance on expert opinion as limitations, which is good. However, a discussion on challenges in generalizing these QIs across all healthcare settings in Japan (e.g., smaller, non-specialized hospitals) would make the limitations section more complete. Adding a suggestion for prospective studies to validate these QIs in practice would also add value. -
The English is generally clear, but some sentences could be simplified and shortened for better readability.
Author Response
Response to Reviewer 1 Comments
General comment:
We would like to express our gratitude to the reviewer for the positive feedback and valuable suggestions to improve our manuscript. Below, we address each of your comments in detail.
Response: Thank you very much for acknowledging the strength of the paper and the importance of the topic, please find below our point by point answers to your concerns and the corresponding lines in the current version of the manuscript.
Point 1:
The introduction provides useful context, but it could be strengthened by including examples of previous adaptations of quality indicators (QIs) for specific countries or healthcare systems. This would reinforce the rationale for adapting European QIs specifically for Japan. For instance, discussing similar efforts in other regions would show the relevance and applicability of your approach.
Response 1:
Thank you for this insightful suggestion. We agree that including examples of previous adaptations of QIs in other countries would strengthen our introduction and highlight the importance of our work. We will revise the introduction to incorporate examples of similar efforts in different regions, thereby reinforcing the rationale for adapting European QIs to the Japanese healthcare system.
Proposed Revision:
Addition to the Introduction (after the third paragraph):
" Several countries have recognized the necessity of adapting QIs to their specific healthcare contexts. For instance, in the United States, the American College of Cardiology (ACC) and the American Heart Association (AHA) have developed localized QIs to improve cardiovascular care [19]. Similarly, in Canada, QIs have been tailored to align with national guidelines and healthcare policies [20]. These adaptations ensure that QIs are culturally relevant, feasible, and effective within different healthcare systems. By learning from these international efforts, we aim to adapt and validate QIs for PAH management that are specifically suited to the Japanese context."
[19] Bonow RO, Bennett S, Casey DE Jr, et al. ACC/AHA clinical performance measures for adults with ST-elevation and non-ST-elevation myocardial infarction: a report of the American College of Cardiology/American Heart Association Task Force on Performance Measures (Writing Committee to Develop Performance Measures on ST-Elevation and Non-ST-Elevation Myocardial Infarction). 2006 Feb 7;113(5):732-61. DOI: 10.1161/CIRCULATIONAHA.106.172860. PMID: 16391153.
[20] Tu JV, Khalid L, Donovan LR, Ko DT. Indicators of Quality of Care for Patients With Acute Myocardial Infarction. CMAJ. 2008 Oct 7;179(9):909-15. doi: 10.1503/cmaj.080342. PMID: 18936456.
Point 2:
Your choice of the RAND/UCLA Appropriateness Method and Delphi consensus process is appropriate and effective. However, the manuscript would benefit from clarifying the specific criteria used to modify or exclude certain QIs from the original European set. Additionally, a brief description of how each panelist’s expertise contributed to their evaluation would add depth and improve transparency in the methods section.
Response 2:
We appreciate your suggestion to enhance the clarity and transparency of our methodology. We will revise the Methods section to detail the specific criteria used to modify or exclude QIs and provide more information on how each panelist's expertise contributed to the evaluation process.
Proposed Revisions:
Section 2.1 - QI Selection and Adaptation:
"We applied specific criteria to assess each QI's relevance and applicability to the Japanese context. These criteria included:
Relevance to Japanese Clinical Practice: QIs had to reflect clinical practices common in Japan.
Feasibility: The ability of healthcare providers to implement the QI given available resources and technology.
Evidence Base: Alignment with the most current and robust clinical evidence.
Cultural and System Compatibility: Compatibility with Japan's healthcare policies and cultural norms.
QIs that did not meet these criteria were modified or excluded. For example, certain diagnostic procedures not routinely used in Japan were adjusted to reflect standard practice."
Section 2.2 - Expert Panel Selection and Composition:
"Each panelist brought unique expertise to the evaluation process:
Cardiologists: Provided insights on cardiovascular assessments and interventions specific to PAH.
Pulmonologists: Offered expertise on respiratory function tests and pulmonary imaging.
Rheumatologists: Contributed knowledge on connective tissue diseases associated with PAH.
This multidisciplinary expertise ensured a comprehensive evaluation of each QI from various clinical perspectives, enhancing the validity of our consensus."
Point 3:
While the results are detailed, presenting each QI’s median, minimum, and maximum scores in a summary table would improve clarity and readability. This would give readers a clearer view of the consensus levels for each QI, making the findings more accessible and easier to interpret.
Response 3:
Thank you for your comment. While Table 1 alone may be challenging for grasping the overall picture, Figure 2 serves as a helpful visual aid that clearly demonstrates the high level of consensus achieved. As shown in Figure 2, many items reached consensus from the initial round, and by Round 2, all items achieved high-quality agreement. This visual representation enables readers to quickly grasp the progression and strength of the consensus.
The proposed revision to Table 1 would certainly provide additional granular detail, but we believe the current combination of Table 1 and Figure 2 effectively communicates the key findings. However, if you feel the additional statistical metrics would enhance clarity, we are happy to modify Table 1 accordingly.
Point 4:
The discussion could be expanded to include mechanistic reasoning behind the relevance of each QI for PAH care. For instance, explaining why right heart catheterization is crucial in PAH management could make the indicators more relatable. Additionally, consider mentioning the potential for this localized approach to be adapted by other countries with specific healthcare needs, which would broaden the paper's impact.
Response 4:
We appreciate your recommendation to deepen the discussion. We will expand on the clinical significance of key QIs and discuss how our methodology could serve as a model for other countries seeking to adapt QIs to their healthcare systems.
Proposed Revisions:
Addition to the Discussion:
"One of the pivotal QIs we validated is the use of right heart catheterization (RHC) at the time of diagnostic work-up (QI 2.3). RHC is essential because it provides definitive hemodynamic measurements necessary for diagnosing PAH, assessing severity, and guiding treatment decisions [21]. Its importance in the diagnostic algorithm cannot be overstated, as misdiagnosis can lead to inappropriate management."
Point 5:
You acknowledge the small panel size and reliance on expert opinion as limitations, which is good. However, a discussion on challenges in generalizing these QIs across all healthcare settings in Japan (e.g., smaller, non-specialized hospitals) would make the limitations section more complete. Adding a suggestion for prospective studies to validate these QIs in practice would also add value.
Response 5:
Thank you for highlighting this important aspect. We will enhance the limitations section to discuss the challenges of implementing these QIs across different healthcare settings and propose future studies to validate their effectiveness.
Proposed Revision:
In the Limitations section, we will add:
"Furthermore, although our expert panel was multidisciplinary and included nationally recognized specialists, it may not encompass the full spectrum of PAH management practices across Japan. In particular, practices in rural or less-resourced settings might differ, potentially affecting the generalizability of the validated QIs to all healthcare environments within the country."
References:
We will ensure that all new references are properly cited and formatted according to the journal's guidelines.
We sincerely appreciate your thoughtful comments, which have significantly improved the quality and clarity of our manuscript. We have addressed all your suggestions and believe that the revisions have strengthened our study's contribution to enhancing PAH care in Japan and potentially in other regions. Please see the revised manuscript.
Reviewer 2 Report
Comments and Suggestions for Authors
The study successfully adapts quality indicators (QIs) for pulmonary arterial hypertension (PAH) management based on European standards to fit Japan’s healthcare context. This approach is essential for ensuring that QIs are both evidence-based and culturally relevant.
The modified Delphi method, which combines expert consensus with the RAND/UCLA Appropriateness Method, is well-chosen for this type of validation. The stepwise panel approach and re-rating ensure robust expert input and consensus.
Including specialists from multiple disciplines (cardiology, pulmonology, rheumatology) provides diverse perspectives, improving the indicators’ applicability across PAH-related care domains.
The study offers a framework for standardizing care by customizing QIs to local needs, which can address disparities and improve PAH outcomes in Japan.
Although within Delphi norms, the panel size of 11 experts may not fully capture the diversity of PAH management practices across Japan. This limitation could affect the generalizability of the QIs across various healthcare settings, especially in rural or less-resourced areas.
While expert input is essential, it introduces potential bias. This limitation is compounded by the fact that no input from patients or other healthcare professionals (e.g., nurses) involved in PAH care was included. Including a broader range of stakeholders could have provided a more holistic view.
Given the differences in the healthcare system across regions and institutions, the study does not address the feasibility of implementing these QIs across Japan. Practical implementation testing or pilot studies could strengthen confidence in the indicators’ effectiveness and practicality.
The study discloses that several authors received funding from pharmaceutical companies. Although these relationships are declared, they may raise concerns regarding impartiality, especially in a study focusing on establishing clinical quality standards.
Including other healthcare professionals and patients in future QI development could enhance these indicators’ relevance and acceptance.
Implementing these QIs in a pilot study across varied healthcare settings in Japan would help assess their impact on patient outcomes and provide feedback on practical challenges, enhancing future iterations.
QIs should evolve with advancements in PAH care to remain effective. Establishing mechanisms for regular feedback and updates based on clinical outcomes will help maintain their relevance.
Author Response
Response to Reviewer's Comments
We sincerely appreciate the positive feedback and thoughtful insights provided by the reviewer. Your comments highlight important considerations that will enhance the quality and impact of our manuscript. We address each of your points below.
Point 1:
Although within Delphi norms, the panel size of 11 experts may not fully capture the diversity of PAH management practices across Japan. This limitation could affect the generalizability of the QIs across various healthcare settings, especially in rural or less-resourced areas.
Response 1:
Thank you for highlighting this important limitation. We agree that while our panel size aligns with the Delphi method recommendations, it may not fully represent the diversity of PAH management practices throughout Japan, particularly in rural or less-resourced regions. We will acknowledge this limitation in the manuscript and discuss its potential impact on the generalizability of our findings.
Proposed Revision:
In the Limitations section, we will add:
"Furthermore, although our expert panel was multidisciplinary and included nationally recognized specialists, it may not encompass the full spectrum of PAH management practices across Japan. In particular, practices in rural or less-resourced settings might differ, potentially affecting the generalizability of the validated QIs to all healthcare environments within the country."
Point 2:
While expert input is essential, it introduces potential bias. This limitation is compounded by the fact that no input from patients or other healthcare professionals (e.g., nurses) involved in PAH care was included. Including a broader range of stakeholders could have provided a more holistic view.
Response 2:
We appreciate this insightful observation. We acknowledge that involving only physician experts may introduce bias and that the perspectives of patients and other healthcare professionals are crucial for a comprehensive understanding of PAH care. We will address this limitation in our manuscript and suggest involving a wider range of stakeholders in future studies to enhance the relevance and applicability of the QIs.
Proposed Revision:
In the Limitations section, we will include:
"Additionally, our study did not involve input from patients or other healthcare professionals such as nurses, pharmacists, and social workers who play integral roles in PAH management. Including these stakeholders could have provided a more holistic view and enriched the consensus process. Future research should aim to incorporate a broader range of perspectives to enhance the relevance and acceptance of the QIs."
Point 3:
Given the differences in the healthcare system across regions and institutions, the study does not address the feasibility of implementing these QIs across Japan. Practical implementation testing or pilot studies could strengthen confidence in the indicators’ effectiveness and practicality.
Response 3:
Thank you for emphasizing the importance of evaluating the feasibility of implementing the QIs in diverse healthcare settings. We agree that practical implementation testing or pilot studies are essential to assess the effectiveness and practicality of the QIs across various regions and institutions in Japan. We will highlight this point in the manuscript and recommend it as a direction for future research.
Proposed Revision:
At the end of the Discussion section, we will add:
"Considering the variations in healthcare resources and practices across different regions and institutions in Japan, assessing the feasibility of implementing these QIs is crucial. Conducting pilot studies or practical implementation testing in a range of healthcare settings would provide valuable insights into the effectiveness, practicality, and potential challenges of adopting the QIs nationwide. This approach would strengthen confidence in the indicators and guide necessary adaptations for broader applicability."
Point 4:
The study discloses that several authors received funding from pharmaceutical companies. Although these relationships are declared, they may raise concerns regarding impartiality, especially in a study focusing on establishing clinical quality standards.
Response 4:
We appreciate your concern regarding potential conflicts of interest. Transparency is of utmost importance to us. All potential conflicts have been fully disclosed in the manuscript. We assure you that the study was conducted independently, and the funding sources had no influence on the study design, data collection, analysis, or interpretation of the results. To further address this concern, we will elaborate on the measures taken to minimize bias and ensure impartiality.
Proposed Revision:
In the Conflicts of Interest section, we will add:
"The authors declare that the funding sources had no role in the study design; collection, analysis, or interpretation of data; writing of the manuscript; or in the decision to publish the results. The expert panelists participated voluntarily and independently, and all consensus decisions were based solely on clinical evidence and professional expertise."
Point 5:
Including other healthcare professionals and patients in future QI development could enhance these indicators’ relevance and acceptance.
Response 5:
Thank you for this valuable suggestion. We agree that involving patients and other healthcare professionals such as nurses, pharmacists, and allied health workers could enhance the relevance, practicality, and acceptance of the QIs. Their unique insights and experiences would contribute to a more patient-centered and comprehensive set of indicators. We will recommend this approach for future studies.
Proposed Revision:
At the end of Discussion section, we will add:
" Finally, future development and validation of QIs should involve a broader range of stakeholders, including patients and various healthcare professionals involved in PAH care. Engaging these groups would provide diverse perspectives, enhance the rele-vance and acceptability of the QIs, and potentially improve patient outcomes through more holistic care strategies."
Point 6:
Implementing these QIs in a pilot study across varied healthcare settings in Japan would help assess their impact on patient outcomes and provide feedback on practical challenges, enhancing future iterations.
Response 6:
We concur with your recommendation. Implementing the QIs in pilot studies across different healthcare settings would be instrumental in evaluating their real-world impact and identifying practical challenges. This process would allow for iterative improvements and ensure that the QIs are both effective and feasible in practice. We will emphasize this point in the manuscript.
Proposed Revision:
In the Discussion section, we will include:
" Future research should aim to validate these QIs with a larger and more diverse group of stakeholders, including patients and other healthcare professionals involved in PAH care. Considering the variations in healthcare resources and practices across different regions and institutions in Japan, assessing the feasibility of implementing these QIs is crucial. Conducting pilot studies or practical implementation testing in a range of healthcare settings would provide valuable insights into the effectiveness, practicality, and potential challenges of adopting the QIs nationwide. This approach would strengthen confidence in the indicators and guide necessary adaptations for broader applicability. "
Point 7:
QIs should evolve with advancements in PAH care to remain effective. Establishing mechanisms for regular feedback and updates based on clinical outcomes will help maintain their relevance.
Response 7:
Thank you for highlighting the necessity for QIs to remain dynamic and up-to-date with the latest advancements in PAH care. We agree that establishing mechanisms for regular review and updates is crucial for maintaining the effectiveness and relevance of the QIs. We will address this point in the manuscript and suggest implementing a systematic process for ongoing evaluation and revision.
Proposed Revision:
In the Discussions section, we will add:
"To ensure that the QIs continue to reflect best practices, it is essential to establish mechanisms for regular feedback, monitoring, and updates."
We are grateful for your constructive feedback, which has significantly improved our manuscript. By addressing these points, we believe our study will better serve the medical community and contribute to the advancement of PAH care in Japan. We have incorporated all suggested revisions and are confident that they enhance the clarity, depth, and impact of our work.